# EyeDAS: Securing Perception of Autonomous Cars Against the Stereoblindness Syndrome

## Abstract

The ability to detect whether an object is a 2D or 3D object is extremely important in autonomous driving, since a detection error can have life-threatening consequences, endangering the safety of the driver, passengers, pedestrians, and others on the road. Methods proposed to distinguish between 2 and 3D objects (e.g., liveness detection methods) are not suitable for autonomous driving, because they are object dependent or do not consider the constraints associated with autonomous driving (e.g., the need for real-time decision-making while the vehicle is moving). In this paper, we present *EyeDAS*, a novel few-shot learning-based method aimed at securing an object detector (OD) against the threat posed by the stereoblindness syndrome (i.e., the inability to distinguish between 2D and 3D objects). We evaluate *EyeDAS*'s real-time performance using 2,000 objects extracted from seven YouTube video recordings of street views taken by a dash cam from the driver's seat perspective. When applying *EyeDAS* to seven state-of-the-art ODs as a countermeasure, *EyeDAS* was able to reduce the 2D misclassification rate from 71.42-100% to 2.4% with a 3D misclassification rate of 0% (TPR of 1.0). Also, *EyeDAS* outperforms the baseline method and achieves an AUC of over 0.999.

## 1 Introduction

After years of research and development, automobile technology is rapidly approaching the point at which human drivers can be replaced, as commercial cars are now capable of supporting semi-autonomous driving. To create a reality that consists of commercial semi-autonomous cars, scientists had to develop the computerized driver intelligence required to: (1) continuously create a virtual perception of the physical surroundings (e.g., detect pedestrians, road signs, cars, etc.), (2) make decisions, and (3) perform the corresponding action (e.g., notify the driver, turn the wheel, stop the car).

While computerized driver intelligence brought semi-autonomous driving to new heights in terms of safety (1), recent incidents have shown that semi-autonomous cars suffer from the stereoblindness syndrome: they react to 2D objects as if they were 3D objects due to their inability to distinguish between these two types of objects. This fact threatens autonomous car safety, because a 2D object (e.g., an image of a car, dog, person) in a nearby advertisement that is misdetected as a real object can trigger a reaction from a semi-autonomous car (e.g., cause it to stop in the middle of the road), as shown in Fig. 1. Such undesired reactions may endanger drivers, passengers, and nearby pedestrians as well. As a result, there is a need to secure semi-autonomous cars against the perceptual challenge caused by the stereoblindness syndrome.

The perceptual challenge caused by the stereoblindness syndrome stems from object detectors' (which obtain data from cars' video cameras) misclassification of 2D objects. One might argue that the stereoblindness syndrome can be addressed by adopting a sensor fusion approach: by cross-correlating data from the video cameras with data obtained by sensors aimed at detecting depth (e.g., ultrasonic sensors, radar). However, due to safety concerns, a "safety first" policy is implemented in autonomous vehicles, which causes them to consider a detected object as a real object even when it is detected by a single sensor without additional validation from another sensor (2; 3). This is also demonstrated in Fig. 1 which shows how

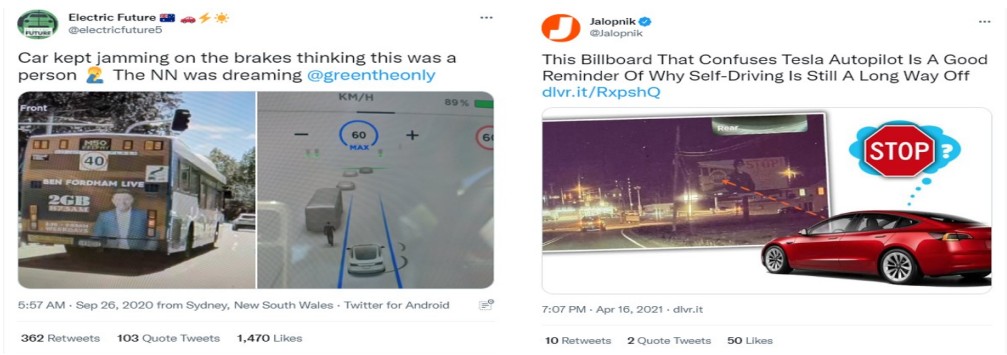

Figure 1: Two well-known incidents that demonstrate how Teslas misdetect 2D objects as real objects.

a Tesla's autopilot triggers a sudden stop due to the misdetection of a 2D object as a real object, despite the fact that Teslas are equipped with radar, a set of ultrasonic sensors, and a set of front-facing video cameras.

In addition, while various methods have used liveness detection algorithms to detect whether an object is 2D/3D (4; 5; 6), the proposed methods do not provide the functionality required to distinguish between 2D/3D objects in an autonomous driving setup, because they are object dependent (they cannot generalize between different objects, e.g., cars and pedestrians) and do not take into account the real-time constraints associated with autonomous driving. As a result, there is a need for dedicated functionality that validates the detections of video camera based object detectors and considers the constraints of autonomous driving.

In this paper, we present *EyeDAS*, a committee of models that validates objects detected by the on-board object detector. *EyeDAS* aims to secure a single channel object detector that obtains data from a video camera and provides a solution to the stereoblindness syndrome, i.e., distinguishes between 2 and 3D objects, while taking the constraints of autonomous driving (both safety and real-time constraints) into account. *EyeDAS* can be deployed on existing advanced driver-assistance systems (ADASs) without the need for additional sensors.

*EyeDAS* is based on few-shot learning and consists of four lightweight unsupervised models, each of which utilizes a unique feature extraction method and outputs a 3D confidence score. Finally, a meta-classifier uses the output of the four models to determine whether the given object is a 2 or 3D object.

We evaluate *EyeDAS* using a dataset collected from seven YouTube video recordings of street views taken by a dash cam from the driver's seat perspective; the 2D objects in the dataset were extracted from various billboards that appear in the videos. When applying *EyeDAS* to seven state-of-the-art ODs as a countermeasure, *EyeDAS* was able to reduce the 2D misclassification rate from 71.42-100% to 2.4% with a 3D misclassification rate of 0% (TPR of 1.0). We also show that *EyeDAS* outperforms the baseline method and achieves an AUC of over 0.999.

In this research we make the following contributions: (1) we present a practical method for securing object detectors against the stereoblindness syndrome that meets the constraints of autonomous driving (safety and real-time constraints), and (2) we show that the method can be applied using few-shot learning, can be used to detect whether an inanimate object is a 2D or 3D object (i.e., distinguishes between a real car from an advertisement containing an image of a car), and can generalize to different types of objects and between cities.

The remainder of this paper is structured as follows: In Section 2, we review related work. In Section 3, we present *EyeDAS*, explain its architecture, design considerations, and each expert in the committee of models. In Section 4, we evaluate *EyeDAS*'s performance under the constraints of autonomous driving, based on various YouTube video recordings taken by a dash cam from several places around the world. In Section 5 we discuss the limitations of *EyeDAS*, and in Section 6, we present a summary.

## 2 Related Work

The ability to detect whether an object is a 2D or 3D object is extremely important in autonomous driving, since a detection error can have life-threatening consequences, endangering the safety of the driver, passengers, pedestrians, and others on the road. Without this capability, Tesla's autopilot was unintentionally triggered, causing the car to: (1) continuously slam on the brakes in response to a print advertisement containing a picture of a person that appeared on a bus (7), and (2) stop in response to a billboard advertisement that contained a stop sign (8). Moreover, attackers can exploit the absence of this capability and intentionally trigger: (1) Tesla's autopilot to suddenly stop the car in the middle of a road in response to a stop sign embedded in an advertisement on a digital billboard (2), and (2) Mobileye 630 to issue false notifications regarding a projected road sign (9).

The need to detect whether an object is 2 or 3D is also important for authentication systems (e.g., face recognition systems) where the identity of a user can be spoofed using a printed picture of the user. Various methods have been suggested for liveness detection (4; 5; 6), however the two primary disadvantages of the proposed methods are that they: (1) fail to generalize to other objects (e.g., distinguish between a real car and a picture of car), since they mainly rely on dedicated features associated with humans (4) (e.g., eye movements (5), facial vein map (6)), which makes them object dependent; or (2) have high false negative rates for pictures of objects that were not taken from a specific orientation, angle, or position (e.g., they fail to detect liveness if the picture of the person was taken from the back). As a result, these methods are not suitable for autonomous driving.

## 3 *EyeDAS*

The requirements for a method used to secure the perception of autonomous cars against stereoblindness syndrome are as follows. The method must be capable of: (1) operating under the constraints of autonomous driving, and (2) securing an object detector that obtains data from a single video camera, because a few commercial ADASs, including Mobileye 630 PRO, rely on a single video camera without any additional sensors, and (3) utilizing just a small amount of training data; the fact that there may be just a small amount of 2D objects in each geographical area necessitates a method with high performance and minimum training so that it can be generalized to different types of objects and between geographical locations.

### 3.1 Architecture

Fig. 2 provides an overview of *EyeDAS*; whenever an object is detected by the vehicle's image recognition model, it is tracked during $t$ consecutive frames sampled at a frequency of $f$ frames per second (FPS), cropped from each frame and serially passed to *EyeDAS*. *EyeDAS* then predicts whether the object is a 2D (e.g., an image of a person) or 3D object (e.g., a real person).

Let $x = (x^1, ..., x^{t-1}, x^t)$ be a time series of $t$ identical RGB objects cropped from $t$ consecutive frames where each object is centered. To predict whether an object is 2D or 3D, we could build a supervised machine learning model which receives $x$, which consists of images of an object to classify, and predicts whether the object detected is 2D or 3D. However, such an approach would make the machine learning model reliant on specific features and thus would not generalize to objects extracted when the vehicle is traveling in different locations or at different speeds, or when the vehicle is approaching the object from different angles or distances.

To avoid this bias, we utilize the committee of experts approach used in machine learning applications (10), in which there is an ensemble of models, each of which has a different perspective of interpreting the incoming data. By combining different perspectives, we (1) create a more resilient classifier that performs well even in cases where one aspect fails to capture the evidence, and (2) reduce the false alarm rate by focusing the classifier on just the relevant input features; *EyeDAS* consists of an ensemble of unsupervised models (experts),

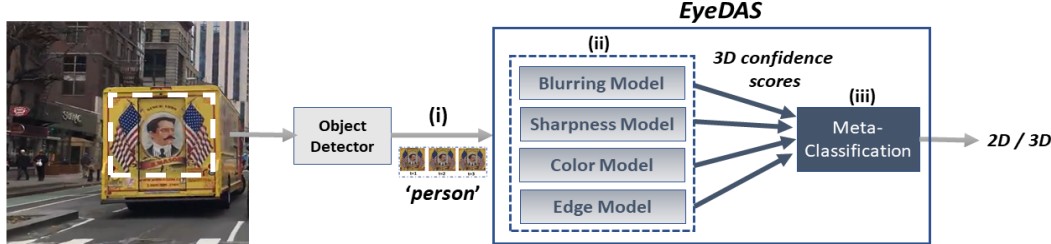

Figure 2: *EyeDAS*'s architecture. When an object is detected, (i) a time series of the cropped object images is transferred to *EyeDAS*, (ii) four types of unique features are processed by four unsupervised models (i.e., experts), resulting in four 3D confidence scores, and (iii) the meta-classifier model interprets the confidence scores and makes the final decision regarding the object (2D or 3D).

each of which outputs a 3D confidence score, and a supervised model (meta-classifier), which produces the final outcome (decision) given the set of confidence scores.

Although each of the proposed unsupervised models (experts) focuses on a different perspective, they have a common property: given $x$, which consists of images of an object to classify as 2D or 3D, each model measures a *difference* between each two consecutive elements in the series; in this study, we show that the combination of a proper feature extraction method together with the proper *distance* metric (applied between each two consecutive elements) is a good basis for building such a classifier. In addition, decisions based on a *distance* observed between two consecutive elements in a given series allows *EyeDAS* to generalize; this approach minimizes dependency on object types or geographical locations. In addition, *EyeDAS* finds the optimal balances between time to decision and classification accuracy; we compare *EyeDAS* utilizing $t > 1$ object frames to a state-of-the-art image classifier that is designed to process a single frame at a time.

From the perspective of a software module like *EyeDAS*, a 3D object itself is not expected to *change* significantly within a short period of time, even in cases in which the 3D object is a human, animal, or vehicle (i.e., a real object). Therefore, it is not trivial to find distance metrics that can detect statistically significant differences that: (1) allow accurate differentiation between 2D and 3D objects by considering just the object, and (2) can be computed quickly. Therefore, we suggest a time-efficient approach that considers objects entangled with their close surrounding background.

Each of the proposed unsupervised models utilizes a unique feature extraction method and a corresponding *distance* metric; these models are designed to process image time series of any size $t$ ($t > 1$). This property is crucial, since the exact time point at which there is a significant difference between two images (in cases in which the object detected is 3D) is unpredictable. In addition, the images to analyze should be represented in such a way that ensures that a statistically significant difference can be efficiently obtained for 3D objects, while the error rate for 2D objects is minimized. In other words, considering the objects entangled with their close surrounding background, we are interested in feature extraction methods whose outcomes for 2D objects and 3D objects are statistically distinguishable within a short period of time.

## 3.2 Proposed Models

Our committee consists of four unsupervised models, each focusing on a different perspective (see Fig. 3 for a demonstration of each model's perspective); each model receives the time series of consecutive images of an object to classify, extracts features from each image, measures a type of *difference* between the features extracted from two consecutive images, and finally outputs a 3D confidence score. Additional details on the four models are provided below:

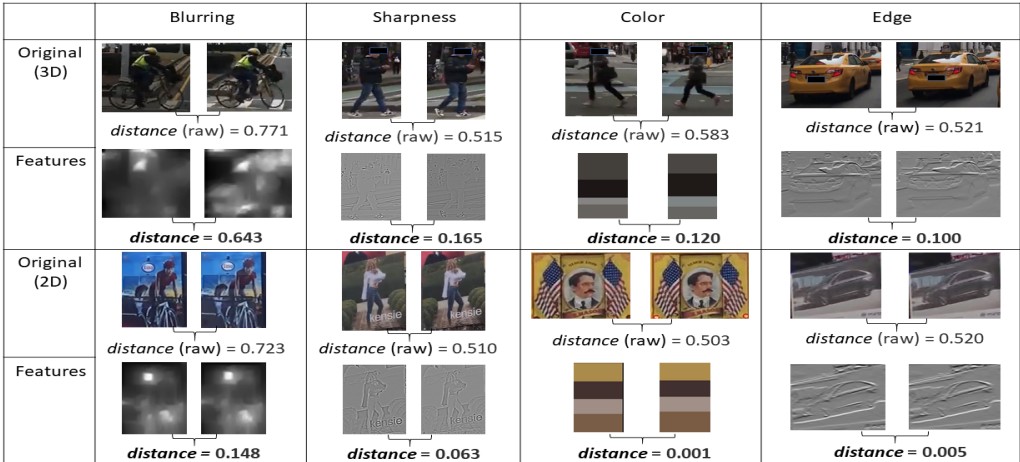

Figure 3: Examples of *EyeDAS* models' views on 2D and 3D objects and the corresponding *distances* (both raw and *EyeDAS*'s distances are presented) calculated as 3D confidence scores.

**Blurring Model (B) -** This model utilizes the automatic focus (auto-focus) capability of video cameras commonly used to build 3D cameras (11). Unlike 2D object images, for 3D object images, the blurring effect differs and is applied alternately on the object and its surrounding background during the auto-focus process; this is reflected in a large amount of contrast, which is observed when examining the **blurring maps** (12) corresponding to the raw images. Thus, using the structural image similarity measure proposed by Wang et al. (13), the blurring model outputs the maximum value obtained by calculating the differences between each two consecutive blurring maps.

**Sharpness Model (S) -** This model utilizes the possible image sharpness-level instability observed in 3D object images due to the objects' movements; the sharpness model extracts the overall **sharpness** level (a numeric value) from each raw image received, using the method described by Bansal et al. (14), and outputs the maximum value obtained by calculating the differences between each two consecutive sharpness levels.

**Color Model (C) -** This model utilizes the possible movement expected in the surrounding environment of 3D objects; this movement is reflected in a large difference in the **color distributions** of the raw images. Thus, the color model extracts the color distributions from each raw image using a clustering algorithm (15), computes the size of the largest cluster observed, and then outputs the maximum value obtained by calculating differences between each two consecutive elements.

**Edge Model (E) -** This model is based on the Sobel edge detector (16), a gradient-based method for estimating the first order derivatives of the image separately for the horizontal and vertical axes; these derivatives are not expected to change significantly for static objects like 2D objects. The edge model operates similarly to the blurring model except for one thing: given the raw images, the edge model extracts the **edging maps** instead of the blurring maps for comparison; extraction is performed using the method described by Gao et al. (17).

**Meta-Classifier -** To make a prediction as to whether or not an object is a 2D or 3D object, we combine the knowledge of the unsupervised models described above in a final prediction; we utilize a gradient boosting (GB) based binary classifier (18) trained on the outputs of the models. We choose the GB algorithm, since it can capture nonlinear function dependencies; in our case, the above unsupervised models complement each other, and their output scores form a nonlinear relationship.

The reader can use the following link to download the code that implements the method (the link will be added after the review, the project zip file has been already shared during paper submission).

## 4 EVALUATION

The experiments described in this section were designed in recognition of the needs of autonomous driving in the real world; to ensure safety, a solution needs to both accurately distinguish between 2D and 3D objects and make a fast decision.

The experiments are aimed at evaluating: (1) the performance of each expert in the committee, (2) the performance of the entire committee, (3) improvement in the false positive rate of ODs when *EyeDAS* is applied for validation, (4) the practicality of *EyeDAS* in real-time environments (in terms of computational resources and speed), and (5) the ability to generalize between objects (i.e., humans, animals and vehicles) and geographical locations (i.e., cities).

All the experiments described in this section, including the speed and memory benchmark (described in Section 4.4), were conducted on an Intel 2.9 GHz Intel Core i7-10700 and 32GB RAM. The machine's operating system was Windows 10.

### 4.1 EXPERIMENT SETUP

**Dataset.** Since some ADASs (e.g., Mobileye 630 PRO) rely on a single channel camera, all of the data collected for our experiments was obtained from single-channel cameras; we utilized seven YouTube video recordings [1] of street views taken by a dash cam from the driver's seat perspective. The distribution of the dataset represents the real distribution of 2D and 3D objects encountered by autonomous driving in the real world: 2,000 RGB objects (i.e., humans, animals, and vehicles) were extracted from driving view videos taken in seven cities: New York (NY, USA), San Francisco (CA, USA), Dubai (United Arab Emirates), Miami (FL, USA), London (UK), Los Angeles (CA, USA), and George Town (Singapore), of which approximately 95% are 3D objects and 5% are 2D objects extracted from billboards.

The objects were extracted and cropped using the highest performing OD described by Redmon et al. (19); in our experiments, each input instance $x$ associated with an object to classify contains up to five images taken at 200 millisecond intervals, starting at the time point at which an object was detected by the OD. Input instance $x_i$ is labeled as 'True' if the instance represents a 3D object and 'False' if the instance represents a 2D object.

**Training.** We denote $TR_{3D}$ and $TR_{2D}$ as two training sets representing 3D and 2D objects respectively. To avoid an unbalanced training set, we extend $TR_{2D}$ by utilizing known image data augmentation techniques (20) to randomly selected instances from $TR_{2D}$; we apply the rotation technique.

Given the outputs calculated by the unsupervised models for each input instance (i.e., the 3D confidence scores), the final meta-classifier training was performed; the best hyperparameters for training were selected using the grid search algorithm (21) using the random shuffling split method and 10-fold cross-validation. We vary the number of estimators in the set of $\{20, 25, ..., 40\}$ trees, while we change the maximum depth of the trees in the set of $\{2, 3, 4\}$. To select the best set of hyperparameters, we evaluated the meta-classifier's performance in terms of accuracy.

In all of the experiments described in this section: (1) $|TR_{3D}| = 150$ and $|TR_{2D}| = 70$, and (2) a decision was made within 200 milliseconds from the moment the object was detected by the OD (i.e., t = 2).

### 4.2 RESULTS

**Performance.** In Fig. 5, we present the receiver operating characteristic (ROC) plot and the area under the ROC (AUC) for different combinations of the blurring (B), sharpness (S), color (C), and edge (E) models; the combination of all four is our proposed method. The ROC plot shows the true positive rate (TPR) and false positive rate (FPR) for every possible prediction threshold, and the AUC provides an overall performance measure of a classifier (AUC=0.5: random guessing, AUC=1: perfect performance).

---

[1]New York City (USA, NY) San Francisco (USA, CA) Dubai (UAE) Miami (USA, FL) London (UK) Los Angeles (USA, CA) George Town (Singapore)

In our case, there is a critical trade-off that must be considered: the classification threshold. A lower threshold will decrease the FPR but often decrease the TPR as well. In Table 1, we provide the TPR and FPR of the models when the threshold is set at 0.5 and for the threshold value at which the TPR=1. As can be seen, the use of all of the proposed models (B+S+C+E) in combination outperforms all other model combinations.

| | 2D Misclassification Rate | | |
|---|---|---|---|
| | | *EyeDAS* Countermeasure | |
| | Without | With | |
| Object Detector | | Threshold@0.5 | Threshold@[TPR=1] |
| yolov3_darknet53 [19] | 100% | 0.0% | **2.4%** |
| yolov3_darknet53_608 [19] | 78.57% | 0.0% | **2.4%** |
| yolov3_mobilenet_v2 [19] | 71.42% | 0.0% | **2.4%** |
| faster_rcnn_resnet50 [23] | 92.85% | 0.0% | **2.4%** |
| faster_rcnn_resnet_101 [23] | 97.61% | 0.0% | **2.4%** |
| ssd_mobilenet_v2 [24] | 78.57% | 0.0% | **2.4%** |
| ssd_inception_v2 [25] | 88.09% | 0.0% | **2.4%** |

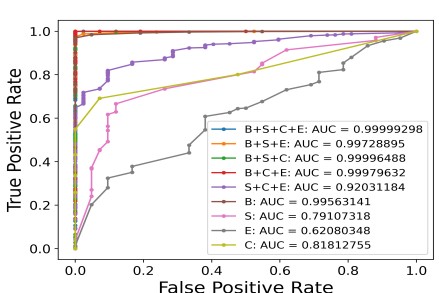

Figure 4: 2D misclassification rates using state-of-the-art object detectors.

Figure 5: The values of the ROC curve and AUC metrics for different model combinations (B: blurring model, S: sharpness model, C: color model, E: edge model).

In Table 2, we compare *EyeDAS*'s performance to that of two other approaches: (1) baseline models based on the state-of-the-art pre-trained image classifiers (i.e., VGG16, VGG19 and Resnet50 (22)); we utilize the known transfer learning technique (23) by re-training these models, and (2) an optimized model similar to *EyeDAS*, except that it is based on a single expert model which considers the raw images as is (i.e., it computes the image similarity based distance (13) between the raw images directly, without extracting any features as *EyeDAS* does). In both cases, 220 instances were randomly selected for training; the distribution of 3D and 2D images is approximately 66.7% and 33.3% respectively, and the data augmentation technique described above was applied to avoid an unbalanced training set.

Each baseline model was re-trained by (1) freezing all the layers except for the output layer which was replaced by two trainable fully connected layers, (2) randomly picking 50 instances from the training set to serve as the validation set, (3) pre-processing the input data (i.e., image resizing and scaling), and finally (4) minimizing the *categorical_crossentropy* loss function on the validation set using the Adam optimizer. The first new layer contained 128 neurons and attached with the relu activation function, and the second layer (output layer) contained two neurons and attached with the softmax activation function. Increasing the first layer's neurons count beyond 128 resulted in poorer results due to overfitting. As can be seen, *EyeDAS* outperforms all of the abovementioned models.

Table 1: *EyeDAS*'s TPR and FPR with different thresholds.

| | | B | S | C | E | B+S | B+C | B+E | S+C | S+E | C+E | B+S+C | B+S+E | S+C+E | B+C+E | *EyeDAS* B+S+C+E |
|---|---|---|---|---|---|---|---|---|---|---|---|---|---|---|---|---|
| Threshold @0.5 | **TPR** | 0.950 | 0.617 | 0.549 | 0.608 | 0.950 | 0.963 | 0.950 | 0.744 | 0.701 | 0.603 | 0.963 | 0.950 | 0.704 | 0.959 | **0.968** |
| | **FPR** | 0.000 | 0.095 | 0.000 | 0.381 | 0.100 | 0.000 | 0.000 | 0.095 | 0.167 | 0.033 | 0.000 | 0.000 | 0.100 | 0.000 | **0.000** |
| Threshold @[TPR=1] | **TPR** | 1.000 | 1.000 | 1.000 | 1.000 | 1.000 | 1.000 | 1.000 | 1.000 | 1.000 | 1.000 | 1.000 | 1.000 | 1.000 | 1.000 | **1.000** |
| | **FPR** | 1.000 | 1.000 | 1.000 | 1.000 | 0.595 | 0.333 | 0.476 | 1.000 | 1.000 | 1.000 | 0.119 | 1.000 | 1.000 | 0.500 | **0.024** |

**TPR**: Ratio of 3D objects detected; **FPR**: Ratio of 2D objects misclassified as 3D objects.

Table 2: Comparison to other approaches.

| | | VGG16 | VGG19 | Resnet50 | Raw | *EyeDAS* |
|---|---|---|---|---|---|---|
| Threshold @0.5 | **TPR** | 0.855 | 0.893 | 0.766 | 0.741 | **0.968** |
| | **FPR** | 0.071 | 0.143 | 0.286 | 0.024 | **0.000** |
| Threshold @[TPR=1] | **TPR** | 1.000 | 1.000 | 1.000 | 1.000 | **1.000** |
| | **FPR** | 0.881 | 0.905 | 1.000 | 1.000 | **0.024** |

**Securing ODs with *EyeDAS*.** To determine how effective *EyeDAS* is as part of a system, we evaluated 2D misclassification rates on seven state-of-the-art ODs (19; 24; 25; 26). The results are presented in Table 4; we present the 2D misclassification rates obtained for each detector before and after applying *EyeDAS* as a countermeasure and the impact of the different thresholds. The results show that for most ODs, when the detector mistakenly classified a 2D object as real (i.e., 3D), *EyeDAS* provided effective mitigation, even for the threshold value at which the TPR=1.

Table 3: Disagreements between the models.

| Disagreement on… | | B, S | B, C | B, E | S, C | S, E | C, E | B, S, C | B, S, E | B, C, E | C, S, E | B, S, C, E |
|---|---|---|---|---|---|---|---|---|---|---|---|---|
| Threshold @0.5 | **2D** | 0.3% | 0.0% | 1.0% | 0.0% | 0.5% | 1.1% | 0.3% | 1.3% | 1.0% | 0.9% | **1.3%** |
| | **3D** | 34.5% | 44.7% | 20.7% | 37.2% | 33.5% | 26.6% | 64.1% | 50.7% | 55.0% | 47.6% | **72.8%** |
| Threshold @[TPR=1] | **2D** | 0.0% | 0.0% | 0.0% | 0.0% | 0.0% | 0.0% | 0.3% | 0.0% | 0.7% | 0.0% | **1.3%** |
| | **3D** | 38.2% | 47.8% | 24.3% | 43.1% | 44.2% | 42.7% | 67.6% | 55.4% | 58.7% | 67.4% | **75.9%** |

**Ablation study.** In order for the committee of experts approach to be effective, there must be some disagreements between the models; although there is a clear indication that each model (i.e., blurring, sharpness, color, and edge) contributes a unique perspective on the final prediction, we perform an ablation study to provide further evidence that each model has a unique contribution. For that, we utilize the SHAP (SHapley Additive exPlanations) framework to explain the model's predictions; each input feature is assigned a score (i.e., a Shapley value) which represents its contribution to the model's outcome. In our case, we are interested in the contributions of each expert (i.e., each input feature to our meta-classifier) to the final prediction. In Table 3, we present the measure of disagreement between each possible sub-committee of experts: given an input instance $x$, we say that a sub-committee disagrees on $x$ if there is at least one expert whose Shapley value is negative and at least one expert whose Shapley value is positive. Interestingly, the full combination of experts (B+S+C+E) has the highest number of disagreements and results in the lowest FPR at TPR=1.

Table 4: Evaluation results for different combinations of locations (NY: New York, GT: George Town, MI: Miami, SF: San Francisco, LA: Los Angeles, DU: Dubai, LO: London).

| Trained in… | Tested in… | FPR Threshold @0.5 | TPR Threshold @0.5 | FPR Threshold @[TPR=1] | TPR Threshold @[TPR=1] |
|---|---|---|---|---|---|
| NY+GT | LA | 0.000 | 0.944 | **0.000** | 1.000 |
| | MI | 0.000 | 0.953 | **0.000** | 1.000 |
| | SF | 0.000 | 0.962 | **0.000** | 1.000 |
| | DU | 0.000 | 0.979 | **0.000** | 1.000 |
| | LO | 0.000 | 0.990 | **0.000** | 1.000 |
| GT+MI | NY | 0.000 | 0.962 | **0.154** | 1.000 |
| | LA | 0.000 | 0.958 | **0.000** | 1.000 |
| | SF | 0.000 | 0.962 | **0.000** | 1.000 |
| | DU | 0.000 | 0.973 | **0.000** | 1.000 |
| | LO | 0.000 | 0.990 | **0.000** | 1.000 |

## 4.3 GENERALIZATION

We also evaluate how *EyeDAS*'s approach generalizes to different geographical locations and even different types of objects.

**Generalization to other geographical locations.** To evaluate *EyeDAS*'s geographical location generalization ability, we trained and tested the models on complementary groups of location types (i.e., cities). For training, we took minimum-sized city type combinations in which there are at least 56 2D objects and at least 120 3D objects, since we observed that less than that amount is insufficient for training the models and thus no meaningful

conclusions could be derived. In Table 4, we present the evaluation results obtained for different geographical location (i.e., cities) combinations. As can be seen, *EyeDAS* is not dependent on the geographical location in which the training data was collected, as it is capable of using the knowledge gained during training to distinguish between 2 and 3D objects.

**Generalization to other types of objects.** To evaluate *EyeDAS*'s object type generalization ability, we trained and tested the models on complementary groups of object types. We focused on humans (HU), animals (AN), and vehicles (VE). As previously done, for training we used minimum-sized object type combinations in which there are at least 56 2D objects and at least 120 3D objects. In Table 5, we present the evaluation results obtained for each object type combination. As can be seen, *EyeDAS* is independent of the type of the object appeared in the training set, as it is capable of using the knowledge gained during training to distinguish between 2 and 3D objects.

Table 5: Evaluation results for different combinations of object types.

| Trained on... | Tested on... | FPR Threshold @0.5 | TPR Threshold @0.5 | FPR Threshold @[TPR=1] | TPR Threshold @[TPR=1] |
|---|---|---|---|---|---|
| Animals + Vehicles | Humans | 0.016 | 0.934 | **0.072** | **1.000** |
| Humans | Vehicles + Animals | 0.000 | 0.715 | **0.249** | **1.000** |
| Humans + Vehicles | Animals | 0.100 | 1.000 | **0.100** | **1.000** |

## 4.4 Speed and Memory Performance

We performed a speed benchmark experiment to assess *EyeDAS* 's performance. We found that *EyeDAS* took an average of 19 ms to process a single object frame, with a standard deviation of 1 ms. This translates to an approximately 220 ms time to decision from the time the object is detected. We also note that the meta-classifier is relatively small (45 kilo bytes) and only utilizes less than 0.01% of the CPU per object frame.

## 5 Limitations

Despite the high performance achieved, *EyeDAS* has some limitations. First, technical factors may influence the performance of *EyeDAS*. For example, *EyeDAS* may perform poorly when image resolution is low or images are taken in low lighting. However, adequate lighting conditions are expected on roads on which autonomous vehicles can typically drive. Second, for 3D objects, if both the detected object and its close surrounding background are stationary (e.g., the object does not move or its surrounding background does not change), then *EyeDAS* may perform poorly. However, if the vehicle is moving or the camera's auto-focus process is operating, then *EyeDAS* 's errors will likely to decrease significantly even for 3D stationary objects. If the vehicle is not moving, the concern for the safety of passengers does not exist.

## 6 Summary

In this paper, we proposed a novel countermeasure which can be used to secure object detectors (ODs) against the stereoblindness syndrome; this syndrome can have life-threatening consequences, endangering the safety of an autonomous car's driver, passengers, pedestrians, and others on the road.

Designed in recognition of the needs of autonomous driving in the real world, the proposed method is based on few-shot learning, making it very practical in terms of collecting the required training set. As presented in the previous sections, *EyeDAS* outperforms the baseline method and demonstrates excellent performance and specifically its: (1) ability to maintain a zero misclassification rate for 3D objects, (2) ability to improve the performance of ODs, (3) practicality in real-time conditions, and (4) ability to generalize between objects (i.e., humans, animals, and vehicles) and geographical locations (i.e., cities).

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
