# OpenReview forum: "EyeDAS: Securing Perception of Autonomous Cars Against the Stereoblindness Syndrome"
_ICLR.cc/2023/Conference — Submitted to ICLR 2023_

### Official Review · Reviewer_NQv8 · 2022-10-21

**Confidence:** 4
**Correctness:** 3
**Technical Novelty And Significance:** 2
**Empirical Novelty And Significance:** 2
**Recommendation:** 5

**Clarity, Quality, Novelty And Reproducibility:**

Clarity: Good. This paper is well-written and easy to understand.
Quality: Fair. Please see the weakness section.
Novelty: Fair. Please see the weakness section.
Reproducibility: Unsure. Dataset and code are not provided.

**Strength And Weaknesses:**

Strengths:
- The paper is well-written and easy to understand.
- The method is simple, fast, and is shown with good empirical performance.
- The authors also conduct a number of ablation studies to show the generalizability and the effectiveness of each component of the proposed method.

Weaknesses:
- Missing details about the dataset and the labeling process: the experiments use a custom dataset collected on YouTube and I think it would be better to provide more details about it. Specifically,
  - Each of the 7 videos is actually quite long (~an hour), but the resulting train/test datasets only have 2000 RGB objects (line 206). What is the process of extracting the objects? Would about 100 (2000 * 5%) non-3D objects be sufficient for evaluation?
  - What is the process of labeling true 3D objects and false 3D objects?
- About the setting: This paper actually proposes a new application problem: few-shot planar/3D object classification. But I am not entirely convinced by its practical soundness in that:
  - I am not sure the few-shot is necessary for such a setting since planar/3D object classification can be object agnostic and can potentially have large enough data for training. Gradient boosted method can work quite well on few data with good feature engineering, but I am not sure it can be better than transfer learning methods with enough data.
- It would be good to also show some qualitative examples of the classification results.
- I am concerned about the minimal contribution: this paper is about a specific and new setting, and the proposed method is simple (gradient boosted tree with good feature engineering) and not new. Though I think it is good to have a simple method that works well, I think it should also show new ideas/insights/understandings to the machine learning community. I do not see enough such contributions from this paper and it looks more like a good technical report.

Minor:
- Styling of the tables: all tables in the paper are from screenshots and I would suggest the authors follow the ICLR style guide (https://github.com/ICLR/Master-Template/raw/master/iclr2023.zip) to present the tables.
- About the evaluation metric: I got confused initially by the 2D misclassification rate and TPR in Table 1: if I understand correctly, TPR is specifically referred to as the true positive rate of 3D objects, and 2D misclassification rate is computed as (FP / (FP + TN) = FPR). I suggest the authors clarify the metrics in the experiment section, and it feels more natural to me to use more conventional names (https://en.wikipedia.org/wiki/Sensitivity_and_specificity)
- Some of the columns/rows in the tables are redundant, e.g., in table 2 and table 3 the TPR row under Threshold @ [TPR=1], in table 5 and table 6 the TPR Threshold @ [TPR=1] column.
- The right image of Figure 1 might not be the right example: the stop sign is also a planar object. Can it be detected by the proposed method?

**Summary Of The Paper:**

In this paper, the author(s) proposed EyeDAS, a few-shot learning-based method to avoid stereoblindness syndrome for object detection (ie. wrongly detect printed objects on billboard/screen as they were in 3D). The proposed method works as a post-processing step. Once any object detection model detects an object with t (t>1) frames, it selects 4 non-learning-based distance metrics (blurring, sharpness, color and edge) to compute a ``3D confidence score''. It uses a gradient-boosted tree to classify if the object is a 3D object. Experimentally, the authors show that the proposed method can achieve close to perfect classification performance on 7 videos (all recorded by a dash cam from the driver’s seat perspective) collected on YouTube, outperforming deep-learning-based transfer learning methods.

**Summary Of The Review:**

Please refer to the Weaknesses section. I am mostly concerned about the dataset details and the minimal contribution.

---

### Official Review · Reviewer_DucZ · 2022-10-23

**Confidence:** 5
**Correctness:** 1
**Technical Novelty And Significance:** 2
**Empirical Novelty And Significance:** 2
**Recommendation:** 1

**Clarity, Quality, Novelty And Reproducibility:**

The  paper is generally well written, with few  errors to fix.
One such error  is  the  mix-up  between Table-4 and Figure-4.

Even if the proposed  approach is novel, it is not justified  and seems very random... The Blurring model is said  to be  related  to autofocus, but the moving cameras have fixed focus, so why mention focus?. Sharpness, Color and Edge are similarly unjustified. The authors do not provide a single convincing example, real of synthetic, that would convince that these models are justified.



**Details Of Ethics Concerns:**

None.

**Strength And Weaknesses:**

The main weakness  of  the  paper is that the problem is not cast as a 3D computer vision problem,  but a vague 2D recognition  problem.
While  it  is clear that the  problem is related  to  estimating scene planarity from a the  motion of  a monocular camera, the  problem is cast as a purely  2D task.

The proposed network  architecture, relying on four simple  models (Burring/Sharpness/Color/Edge) is not well justified and is  not related to the fundamentals of  the problem. This makes it  impossible  to  see how or why  it  would  work.

The choice of  5  images over a  1 second time lapse is  not  properly  justified,  especially when the  proposed  algorithm uses only 2 images to make a decision. The time interval  should be chosen in accordance to  the motion of the  camera,  to ensure that enough parallax is present.

Comparing this architecture, which uses 2 images, with other  architectures that use only one image (and are trained on a different problem) seems somewhat unfair. But maybe I misunderstood that result (Figure 4).



**Summary Of The Paper:**

This paper is interested in discriminating between a 2d view of a 3D object  and an actual  3D object, in the  context of autonomous  driving.
The goal is  to get rid of false detection of objects that are actually pictures of objects,  thereby increasing navigation safety.

Although the author does not express it this way,  the  problem is essentially to  ascertain if the  scene is planar or not in a monocular sequence of two images.
Although the problem is interesting, it  is  cast  in a very vague,  non geometrical way,  which  results  is  an ad hoc algorithm without justification,  especially with  regards to geometry.


**Summary Of The Review:**

This paper has failed to properly identify the exact 3d  computer vision problem  it  tries to  solve, which  is  scene planarity under motion  stereo. Because of this, the proposed model is unjustified, and there  is no  basis to believe  that  it actually works, or would would work well in practice.

The problem is interesting and  important, and I hope the  authors  will consider it from the angle of 3d computer vision, as a planarity estimation problem. It would be ideal, in my  opinion, to explore the  problem in a synthetic framework first (simulating a camera motion in a scene with objects that  are planar and non  planar), before jumping to real images.

---

### Official Review · Reviewer_KB85 · 2022-10-24

**Confidence:** 3
**Correctness:** 3
**Technical Novelty And Significance:** 3
**Empirical Novelty And Significance:** 2
**Recommendation:** 5

**Clarity, Quality, Novelty And Reproducibility:**

The proposed method is novel and clear. The authors provide the code and the reproducibility is guaranteed.

**Strength And Weaknesses:**

Strength:
--The proposed problem of stereoblindness syndrome. i.e., the inability to distinguish between 2D and 3D objects, is interesting and meaningful.  It’s a practical problem we should solve to improve the robustness of autonomous vehicles.
- The paper is well written and organized. And it’s easy to follow.


Weakness:
- The major problem is the evaluation. Firstly, the whole experiment dataset is only collected from seven YouTube video recordings with hundreds of  annotated objects for training and testing. This experiment benchmark are not convincing enough because of the small data quantity.
- And the baseline result used for comparison is poor and not reasonable.  The baseline model does not converge well with such few data. If trained with enough data, I think the baseline model can achieve comparable performance.
- The proposed method is a few-shot method. But in the driving scenerios,  2D / 3D objects are common. It’s easy to get more annotated data for training the model. Few-shot methods are not practical for the problem of stereoblindness syndrome.
- The proposed method is not lightweight enough. 200 ms latency is not enough for real-time applications.



**Summary Of The Paper:**

This work prorposes a few-shot learning-based method named EyeDAS for securing object detectors against the stereoblindness syndrome (i.e., the inability to distinguish between 2D and 3D objects). It leverages the low-level features of image to solve the problem. Four unsupervised models (for blurring, edge, color and sharpness) repectively predict 3D confidence scores and a meta-classifier interprets the confidence scores and makes the final decision. EyeDAS is evaluated on a dataset collected from seven YouTube video recordings.

**Summary Of The Review:**

The evaluation is based on a self-collected  dataset with too few data samples, which makes the effectiveness of the proposed method less convincing.  And the few-shot method  is with less practical value, because annotated data are easily collected and supervised methods work well.

---

### Official Review · Reviewer_d1dP · 2022-10-25

**Confidence:** 4
**Correctness:** 3
**Technical Novelty And Significance:** 2
**Empirical Novelty And Significance:** 3
**Recommendation:** 3

**Clarity, Quality, Novelty And Reproducibility:**

Since this method does not involve learning the representation used but instead uses hand-crafted features, I would argue that it is not directly relevant to ICLR.  It may be more suited for a computer vision applications conference or workshop.

Since the proposed method uses temporal differences between subsequent input frames, it should be compared to approaches that also have that benefit.  Unfortunately the baseline methods all only use a single image as input.   I believe the hand-crafted features used in this method could easily be learned with an e2e learned approach if the model being trained had subsequent frames as input.  I think taking that approach would improve this paper and make it more relevant to ICLR.



**Strength And Weaknesses:**

Strengths:
* Paper deals with an important problem
* This paper presents a useful insight that considering the appearance of the target object over time can provide a strong cue for this problem
* The paper shows that the technique performs well on the dataset they used.

Weaknesses:
* This paper uses hand-crafted rather than learned features and does not present a convincing case that there is a good reason to do so.
* The baseline methods are straw-men that do not benefit from temporal information as the proposed method does.


**Summary Of The Paper:**

This paper tackles an important problem facing autonomous vehicle perception - how to distinguish between a 3d object and a 2d representation of a 3d object?  A useful insight of this paper is that having information of the target object over time leads to important features that help solve this problem.  The paper proposes a model based on 4 heuristic hand-crafted features together with a gradient boosting meta classifier.

**Summary Of The Review:**

This is paper addresses an important problem and demonstrates on a dataset that the method works with accuracy that may be high enough to solve this problem in a practical setting.  The paper makes the insight that using sequences of images over time can help solve this problem.  Unfortunately it does not use a learned representation to achieve this but rather uses hand-crafted features, so is not directly relevant to ICLR.  The baselines that it uses to compare against did not have the advantage of the temporal sequence of images and therefore are not apples-to-apples comparisons.

---

### Decision · Program_Chairs · 2023-01-20

**Decision:**

Reject

**Justification For Why Not Higher Score:**

Based on the recommendation of the reviewers and the absence of author feedback, all reviewers agreed that the paper tackles an important problem but there are many weaknesses in order to accept the paper.

**Justification For Why Not Lower Score:**

N/A

**Metareview: Summary, Strengths And Weaknesses:**

# Summary
This paper tackles an important problem discriminating between a 2d view of a 3D object and an actual 3D object, in the context of autonomous driving.  Hence, the main objective is to get rid of false detection of objects that are pictures of objects, thereby increasing navigation safety. It leverages the low-level features of image to solve the problem. Four unsupervised models (for blurring, edge, color and sharpness) respectively predict 3D confidence scores and a meta-classifier interprets the confidence scores and makes the final decision. EyeDAS is evaluated on a dataset collected from seven YouTube video recordings.

# Strengths:
- The proposed method is novel and clear.
- Paper deals with an important problem
- This paper presents a useful insight that considering the appearance of the target object over time can provide a strong cue for this problem
- The paper shows that the technique performs well on the dataset they used.
- The paper is well written and organized. And it’s easy to follow.

# Weaknesses:
- The proposed network architecture relies on four hand-crafted simple models (Burring/Sharpness/Color/Edge) than learned features. It does not present a convincing case that there is a good reason to do so; hence it is not well justified and is not related to the fundamentals of the problem.
- The baseline methods do not benefit from temporal information as the proposed method does; hence the comparison should be  comparison is poor and not reasonable. The baseline model does not converge well with such few data. If trained with enough data, the baseline model can achieve comparable performance.
- Missing details about the dataset and the labeling process: the experiments use a custom dataset collected on YouTube and  it would be better to provide more details about it.